# Coordinative Changes in Metabolites in Grape Cells Exposed to Endophytic Fungi and Their Extracts

**DOI:** 10.3390/molecules27175566

**Published:** 2022-08-29

**Authors:** Jin-Zhuo Qu, Fang Liu, Xiao-Xia Pan, Chang-Mei Liao, Tong Li, Han-Bo Zhang, Ming-Zhi Yang

**Affiliations:** 1School of Ecology and Environmental Science, Yunnan University, Kunming 650504, China; 2School of Life Science, Yunnan University, Kunming 650504, China; 3School of Chemistry and Environment, Yunnan Minzu University, Kunming 650504, China

**Keywords:** endophytic fungi, fungal extracts, grape cells, metabolite modification, coordinately responding metabolites (CRMs), differentially responding metabolites (DRMs)

## Abstract

Endophytes and their elicitors can all be utilized in regulating crop biochemical qualities. However, living endophytes and their derived elicitors are always applied separately; little is known about the similarities and differences of their effects. To increase the efficiency of this system when applied in practice, the present work profiled simultaneously the metabolomes in grape cells exposed to endophytic fungi (EF) and their corresponding fungal extracts (CFE). As expected, grape cells exposed separately to different fungi, or to different fungi derived extracts, each exhibited different modifications of metabolite patterns. The metabolic profiles of certain EF- and CFE-exposed grape cells were also differently influenced to certain degrees, owing to the presence of differentially responding metabolites (DRMs). However, the detected majority proportions of coordinately responding metabolites (CRMs) in both the EF- and the CFE-exposed grape cells, as well as the significantly influenced metabolites (SIMs) which are specific to certain fungal strains, clearly indicate coordinative changes in metabolites in grape cells exposed to EF and CFEs. The coordinative changes in metabolites in EF- and CFE-treated grape cells appeared to be fungal strain-dependent. Notably, several of those fungal strain-specific CRMs and DRMs are metabolites and belong to amino acids, lipids, organic acids, phenolic acids, flavonoids, and others, which are major contributors to the biochemistry and sensory qualities of grapes and wines. This research clarifies the detailed responses of metabolites in grape cells exposed to EF and CFEs. It also demonstrates how endophytes can be selectively used in the form of extracts to produce functions as CRMs of the living fungus with increased eco-safety, or separately applied to the living microbes or elicitors to emphasize those effects related to their specifically initiated SIMs and DRMs.

## 1. Introduction

Endophytes are microorganisms which colonize internal sections of plants for all or part of their lifetime, and the nature of plant–endophyte interactions ranges from mutualism to pathogenicity [1]. The study of endophytes within host plants is a topic of great relevance due to their potential for use in an agricultural context. As a result, endophytes that have beneficial interactions with their hosts attract great attention for possible utilization in crop quality management [2,3,4]. The application of these symbiotic microorganisms, particularly in perennial woody crops such as grapevines, can facilitate long-term positive effects for candidate endophytes [5]. Endophytes with positive functions, such as adaptation to abiotic stresses, defense against pathogens, growth promotion, and the improvement of the biochemical qualities of grapevines, have been investigated and developed [5]. These endophytes, such as the bacterial endophyte *Paraburkholderia phytofirmans* strain PsJN, are conducive to the development of a chilling stress-preventive trait in plants bearing this endophyte [6,7]. Diketopiperazines, produced by the endophytic fungi (EF) strain *Alternaria alternate,* are isolated from grapevine leaves and show antifungal activity against the pathogen *Plasmopara viticola*, which causes downy mildew [8]. The inoculation of some EF strains, such as CXB-11 (*Nigrospora sp*.) and CXC-13 (*Fusarium sp*.), aid in reducing sugar, total flavonoids, total phenols, trans-resveratrol, and the activities of phenylalanine ammonia-lyase, in both the leaves and berries of the grapevine [9]. In a dual-culture experiment that included controls with no fungal exposure, EF introduced novel metabolites with certain degrees of fungal strain specificity into grape cells [10]. These results confirmed the role played by endophytes in grapevines, and the prospects of using such endophytes to regulate the quality of grapes and their wines.

Generally, the application of endophytes can be undertaken directly through the living organisms themselves, or indirectly through derived elicitors. Living endophytes are always used when long-term effects on their host plants are desired. However, there are risks associated with this method; the living endophytes may sometimes become pathogens or may even cause unpredictable eco-safety problems. A great deal of endophytes, isolated from healthy plant tissues, have been proven to be conditional disease-causing agents to the host plant or plants of other species [1]. It has also been found that most purposely-introduced endophytic strains in a plant’s phyllosphere or rhizosphere will disappear within a short time, before producing any detectable effects [11]. However, the endophytic extract or elicitor could be designed for use at certain times and in specific dosages, without concerning their pathogenicity and other ecological risks. Therefore, it is necessary to choose appropriate forms of endophytes in practice. To do so, it is necessary to clearly elucidate the similarities and differences between utilizing certain living endophytes and the corresponding elicitors in plants before such systems can be successfully applied in practice.

To date, research concerned with the functions of living endophytic organisms and their derived elicitors on host plants has always been carried out separately, and the effects associated with using different forms of endophytes have rarely been discussed. In this research, three EF strains (C11, R12, and R32) belonging to the genera *Fusarium*, *Alternaria,* and *Niqrospora,* respectively, were chosen to be investigated simultaneously. The metabolic effects of these fungal strains and their soluble extracts on grape cells were studied using a popular method of metabolite profiling. All selected EF strains were previously detected as the dominant distribution in grapevine leaves, and the living fungi are proven to have significant biochemical impacts on grape cells [12]. In addition, many species belonging to these fungal genera are proven to be plant disease-causing pathogens, and the direct use of these living fungi may introduce risks to target crops. 

## 2. Materials and Methods

### 2.1. Preparation of In Vitro Grape Callus

A cell line (CBL) induced from the flesh of grape berries (*Vitis vinifera*, cv. Cabernet sauvignon) was used in this study. A B5 solution with 3% sucrose, 0.2 mg/L cytokinin, 0.1 mg/L naphthylacetic acid (NAA), and 0.8% agar (B5 agar medium) was prepared as the medium for the callus sub-culture, along with subsequent treatments with fungi or fungal extracts. The grape calli that were prepared for the experiment were taken during the logarithmic growth phase.

### 2.2. Preparation of Endophytic Fungi (EF) and the Corresponding Fungal Extracts (CFE)

Three EF strains, C11 (*Fusarium sp*.), R12 (*Niqrospora sphaerica*), and R32 (*Alternaria alternaria*), were used in this experiment. These EF strains were previously isolated from grapevine leaves (*V*. *vinifera* L.) from local vineyards in Yunnan Province, China, using the patch culture method, and they were molecularly identified using ITS DNA sequences [12]. The living fungi that were used to establish the dual culture within grape cells were cultured on potato dextrose agar (PDA) plates for 1 week, and fungal mycelia discs generated with a 0.5 cm punch were used in dual cultivation. To prepare the fungal extracts, fresh fungal mycelia were collected via filtration from suspension cultures. After being washed several times with distilled water, fungal mycelia were precisely weighed and ground into homogenates with 10 times the mycelia weight of distilled water, followed by extraction in an ultrasonic cleaner for 30 min. The mixtures were centrifuged at 8000 rpm for 15 min, and the supernatants were preserved at 4 °C as fungal extract solutions for further use.

### 2.3. Treatment of Grape Cells with Living Fungi and Fungal Extracts

The solid dual-culture systems were established according to Huang et al. [13], with some modifications [12]. Instead of inoculating grape calli into the middle of Petri dishes, they were inoculated into one side of B5 plates, and fungal mycelium discs were inoculated on the opposite side of the plates, 5 days after the inoculation of the grape calli. In the absence of treatment controls, PDA discs of the same size without fungi were inoculated. The dual cultures and controls were cultured continuously under dark conditions in an oven at 25 °C for 5 days. Every pair of treatment and control discs included at least 5 biological replicates. The grape cells were harvested for metabolite profiling. For treatment with fungal extracts, the solutions of extracts (prepared previously and stored at 4°C) were then diluted to concentrations equivalent to 10 µg fresh mycelium per microliter (determined by an earlier trial). The diluted mycelial extract solutions were firstly filter sterilized, and a 0.1 mL aliquot was evenly transferred onto the medium surface of each B5 agar plate. The same volume of sterilized distilled water was added to B5 agar plates as a control. Approximately 1.5 g of prepared grape calli was inoculated into the same place where the treatments were inoculated on each plate and was cultured for 5 days, also in dark conditions in an oven at 25 °C. Grape calli treated with different fungal extracts and controls were harvested and sent for metabolite profiling.

### 2.4. Metabolites Profiling

Metabolite profiling was performed using a popular metabolome profiling method. The freeze-dried samples were extracted as previously described [14]. The extracts were analyzed using an LC-ESI-MS/MS system (UPLC, Shim-pack UFLC SHIMADZU CBM30A system; MS, Applied Biosystems 6500 Q TRAP). Metabolite quantification was performed using a scheduled multiple reaction monitoring (MRM) method, which has been described in detail [14]. Metabolite profiling was performed in 3 biological replicates. The identified metabolites were subjected to orthogonal partial least squares discriminant analysis (OPLS-DA).

### 2.5. Statistical Data Analysis

The significantly influenced metabolites (SIMs) in grape cells of certain treatments were identified by comparing sample groups of treatments and controls. They were determined according to the variable importance in project (VIP) ≥ 1 and absolute Log2FC (fold change) ≥ 1. VIP values were extracted using the OPLS-DA results, which also contained score plots and permutation plots, generated using the R package ropls. The data were log transformed (log2) and pared to scaling before performing the OPLS-DA. To avoid overfitting, a permutation test (200 permutations) was performed. The response index (RI) was used to indicate the quantity of responses of certain metabolites in treatments relative to the control. The RI was calculated using the following formula:

RI = (PA_treatment_ − PA_control_)/PA_control_. PA_treatment_ represents the mean value of the peak area of a certain metabolite in the treatment, and PA_control_ is the mean value of the peak area of the same metabolite in the control. After comparing the RI values of certain metabolites between treatments (always between paired treatments, and a paired treatment was defined as the treatment of living EF and CFEs, such as C11 and C11E), the detected metabolites in certain paired treatments were then categorized as coordinately responding metabolites (CRMs), differentially responding metabolites (DRMs), and other metabolites that have not been obviously influenced by either of the paired treatments. CRMs were defined as the metabolites that were coordinately up- or downregulated in grape cells in the paired treatments, with the absolute RI values of the metabolites in either of the paired treatments greater than or equal to 0.1. Accordingly, DRMs were defined as the metabolites that were oppositely regulated in grape cells in the paired treatments (i.e., the metabolites in grape cells of one treatment were upregulated, while downregulated in grape cells of another treatment),and were determined according to the absolute value of the RI difference in the metabolites between the paired treatments greater than or equal to 0.2. A principal component analysis (PCA), a partial least squares discriminant analysis (PLSDA), and heatmaps with a hierarchical clustering analysis were performed using the web platform MetaboAnalst (https://www.metaboanalyst.ca/ (accessed on 23 April 2021)). Figures were generated using Sigma Plot 12.5 (Systat Software Inc., San Jose, CA, USA), Excel (software), and R.

## 3. Results

### 3.1. Grape Cells Exposed to Different EF and Different EF Derived Extracts Differentially Modified the Metabolite Profiles

In total, the 306 metabolites identified in the grape cells belong to compound classes of amino acids, organic acids, alkaloids, nucleotides, phenolic acids, lipids, flavonoids, terpenoids, quinones, and others (Appendix A). Using a principal component analysis (PCA), the replicates of one treatment were grouped together in the plots, with few exceptions (Figure 1). A mixed sample was randomly uploaded for quality control (QC) during metabolite profiling, and the results of these QC samples were also clustered together and displayed within the sample groups in PCA to indicate the validation of the analyses (Figure 1). In PCA, PC1 resolved treatment R12 from other samples, while PC2 resolved all treatments into two groups (R12, R12E, C11E, and C11 formed one group, and samples from other treatments formed another group) (Figure 1). Samples belonging to one treatment pair (such as C11 and C11E) are closely clustered in the PCA (Figure 1). 

Heatmaps developed with hierarchical clustering analysis categorized the treatments and the detected metabolites in grape cells from diverse treatments (Figure 2 and Appendix A). The biological replicates of one treatment were also clustered together, and the samples that belonged to one pair of treatments, such as R12 and R12E, and R32 and R32E, were close due to their similar metabolic impacts on grape cells (Figure 2 and Appendix A). The treatments C11 and C11E, although they belonged to one pair of treatments, were separately clustered in this assay (Figure 3). With reference to the response patterns, the 50 most influenced metabolites were divided into three clusters (CI, CII, and CIII) and six sub-clusters (CI1, CI2, CI3, CII1, CII2, CIII1, CIII2, and CIII3) (Figure 2). Sub-cluster CI1 included three amino acids (L-ornithine, L-histidine, and L-proline), one organic acid (L-homoserine), and a lipid (pentadecanoic acid). These metabolites were obviously downregulated in C11-, C11E-, and R32-treated grape cells, and were slightly upregulated in R32E-exposed grape cells (Figure 2). Sub-cluster CI2 included seven amino acids (L-leucine, L-aspartic acid, L-phenylalanine, L-valine, L-glutamine, L-(+)-lysine, and L-isoleucine), three alkaloids (6-deoxyfagomine, 4,5,6-trihydroxy-2-cyclohexen-1-ylideneacetonitrile, and *N*-benzylmethylene isomethylamine), one nucleotide and derivatives (6-methylmercaptopurine), and a vitamin (4-pyridoxic acid). These metabolites were mainly downregulated in grape cells exposed to C11, C11E, R12, and R12E (Figure 2). Cluster CII incorporated diverse classes of metabolites, such as lipids, coumarins, quinones, phenolic acids, amino acids, alkaloids, organic acids, and stilbene, which were greatly promoted in C11-exposed grape cells (sub-cluster CII1) and in R12-treated grape cells (sub-cluster CII2) (Figure 2). The metabolites in sub-cluster CIII1 belong to the compound classes of terpenoids, alkaloids, phenolic acids, saccharides, and alcohols and organic acids, and they were downregulated in most fungi and fungal-extract exposed grape cells. They were also slightly promoted in C11E-treated grape cells relative to the control (Figure 2). Sub-cluster CIII2 includes phenolic acids (3-hydroxy-4-isopropylbenzylalcohol 3-glucoside, 3-hydroxy-5-methylphenol-1-oxy-β-D-glucose, and 1′-O-vanilloyl-β-D-glucoside), flavonoids (dihydroquercetin, eriodictyol 7-O-glucoside, and Wistin), nucleotides and derivatives (xanthosine and 9-[β-D-arabinofuranosyl] hypoxanthine), organic acids (aminomalonic acid, terpene [Dihydrocornin], stilbene [resveratrol-O-diglucoside]), and a vitamin (biotin), which were mainly promoted in C11E-, R12-, and R12E-exposed grape cells (Figure 2).

### 3.2. Proportions of SIMs Specific to a Certain Endophytic Fungal Strain Were Co-Initiated in EF- and CFE-Exposed Grape Cells

Comparison with no-treatment controls showed that different counts of significantly influenced metabolites (SIMs) were generated in grape cells treated with EF and CFEs (Figure 3). Regardless of which fungal strain was used, the exposure to living EF tended to initiate more counts of SIMs in grape cells than exposure to CFEs (Figure 3a–c). The detected counts of SIMs in EF- and CFE-exposed grape cells listed from the most to the least were R12 > C11 > R32 and R12E > R32E > C11E, respectively. Both the living organism and the extract of fungal strain R12 produced the largest impact on grape cells, as judged by the triggered counts of SIMs (Figure 3b). In addition, living fungi exposure tended to trigger more upregulated SIMs than CFEs (Figure 3a–c).

Venn plots were used to display specific and co-generated counts of SIMs among or between treatments (Figure 3d–i). Exposure to living fungal strains C11, R12, and R32 caused 16, 20, and eight specific SIMs, respectively, and these SIMs occupied 44.4%, 48.8%, and 30.8%, respectively, of the total counts of the SIMs initiated by the EFs (Figure 3d). Accordingly, the counts (proportions) of specific SIMs in C11E-, R12E-, and R32E-treated grape cells were one (11.1%), 17 (60.7%), and four (28.6%), respectively (Figure 3e). The exposure to three living fungal strains triggered nine co-generated SIMs, which occupied 25.0%, 22.0%, and 34.6%, respectively, of the C11-, R12-, and R32-initiated total SIMs in grape cells (Figure 3d). Similarly, fungal extracts C11E, R12E, and R32E caused seven co-initiated SIMs, which occupied 77.8%, 25.0%, and 50.0%, respectively, of the total SIMs generated in the CFE-treated grape cells (Figure 3e). Notably, major proportions (88.9%, 75%, and 71.4%) of the SIMs in C11E-, R12E-, and R32E-treated grape cells were simultaneously detected in C11-, R12-, and R32-exposed grape cells, respectively (Figure 3g–i). Only five SIMs were co-detected in all grape cells exposed to fungi and fungal extracts (Figure 3f). In addition, exposure to living fungi C11 and R12 generated the most counts of specific SIMs (13 and 12, respectively), while C11E and R32E treatments led to almost no specific SIMs in grape cells (Figure 3f).

In all, 76 SIMs were detected in grape cells exposed to fungi and fungal extracts enriched in 38 KEGG pathways (Table 1 and Appendix A). Five of the SIMs, namely, L-tyramine, *N*-acetylmethionine, camaldulenic acid, 2-hydroxyoleanolic acid, and α-viniferin, were significantly downregulated in the grape cells of all treatments (Table 1). Accordingly, KEGG pathways for protein digestion and absorption, metabolic pathways, biosynthesis of secondary metabolites, tyrosine metabolism, methane metabolism, alkaloid biosynthesis, and ligand-receptor interaction were enriched in all grape cells exposed to EF and fungal extracts (Appendix A). All other SIMs and the enriched KEGG pathways were specifically detected in grape cells exposed to fungi or fungal extracts (Table 1 and Appendix A).

### 3.3. Grape Cells Exposed to EF and CFEs Selectively Influenced Different Classes of Metabolites

Grape cells exposed to fungi and fungal extracts were differently influenced by the metabolites of different classes, as judged by the generated ratios of SIMs (Table 1). More than one-quarter of all detected metabolites in classes of amino acids, lipids, and phenolic acids were SIMs, and half of the detected secondary metabolites belonging to other classes, such as flavonoids, terpenes, and stilbenes, were SIMs. Smaller proportions of SIMs were detected in compound classes of alkaloids, nucleotides and derivatives, and saccharides and alcohols in the grape cells under various treatments (Table 1). The influence of fungi and fungal extracts on grape cellular metabolites displayed a certain degree of fungal-strain specificity. The exposure to fungus C11 and its extracts (C11E) may have influenced the metabolites in the compound class of amino acids and their derivatives, while the exposure to fungi R12 and R12E initiated more SIMs in the class of lipids (including glycerol ester, lysophosphatidylcholine, and lysophosphatidyl ethanolamine) (Table 1). Other classes of metabolites in grape cells were influenced to a greater or lesser degree by EF and CFEs (Table 1). In most cases, the co-regulated metabolites in the EF- and CFE-treated grape cells maintained the same trends of up- or downregulation (Table 1). Many of the SIMs in grape cells treated with fungal extracts were also detected in the CFE-exposed grape cells (Table 1). Notably, the metabolite ε-viniferin was upregulated in grape cells treated with living fungi, while this compound was greatly downregulated in fungal-extract treated grape cells (Table 1). The peptide L-glutaminyl-L-valyl-L-valyl-L-cysteine was significantly promoted (Log2(FC) > 5) in all living fungi and R12E-exposed grape cells, but this peptide was not detected in C11E- and R32E-treated grape cells (Table 1). Piceid, a stilbene class of metabolite, was greatly promoted in all treated grape cells in this experiment (Table 1). 

### 3.4. Coordinative and Differential Responses in Metabolites between EF- and CFE-Exposed Grape Cells

The 60 metabolites with the greatest relative contents (top 60 metabolites), which occupied more than 90% of the relative content of all detected metabolites, were chosen to analyze the coordinated changes in metabolites in living EF- and CFE-exposed grape cells (Figure 4). Coordinated modifications of metabolites in grape cells were observed when grape cells were separately exposed to living EF and CFEs, as evaluated by the response indexes (RIs) (Figure 4). The degree of coordination among metabolites and the species of the coordinately responding metabolites (CRMs) in EF- and CFE-exposed grape cells led to obvious fungal strain-dependence (Figure 4). In all three paired treatments, the metabolic responses of grape cells exposed to R12 and R12E produced the greatest degree of coordination, with the relativity (Pearson) of the top 60 metabolites (R_top60_) being 0.921 (P < 0.01). In contrast, metabolite changes in C11- and C11E-treated grape cells showed the lowest degree of coordination (R_top60_ = 0.530, P < 0.01). The degrees of coordination of metabolites in grape cells exposed to R32 and R32E fell within the ranges of the paired treatments discussed above (R_top60_ = 0.730, P < 0.01) (Figure 4a). Among the top 60 metabolites, 26 CRMs in grape cells exposed to C11 and C11E were detected, with 10 of them coordinately upregulated and 13 coordinately downregulated (Figure 4b). Accordingly, R12 and R12E exposure also triggered 26 CRMs in grape cells, with half of these metabolites coordinately upregulated and another half downregulated. R32 and R32E treatments detected the least counts (14) of CRMs. Among these CRMs, four metabolites (piceid, lysoPC 18:1, γ-linolenic acid, and uridine 5′-diphospho-D-glucose) were detected in grape cells of all of the paired treatments. Piceid and lysoPC 18:1 were coordinately upregulated, and γ-linolenic acid and uridine 5′-diphospho-D-glucose were coordinately downregulated (Figure 4b). Other CRMs were detected in one or two paired treatments, and exhibited obvious fungal-strain specificities (Figure 4b).

In addition to the CRMs, differentially responding metabolites (DRMs) were also detected in living EF- and CFE-exposed grape cells, and the number and species of DRMs appeared in fungal-strain specificity (Figure 4b). R32- and R32E-exposed grape cells detected the highest number of DRMs (11), followed by C11- and C11E-treated grape cells (9). The exposure of R12 and R12E only triggered two DRMs in grape cells (Figure 4b).

## 4. Discussion

### 4.1. Grape Cells Exposed to Different Fungi and Different Fungal Derived Extracts Differentially Modified the Metabolite Profiles

The beneficial metabolic interactions between endophytes and their host plants are attracting attention for their potential role in regulating the biochemical qualities of crops [5,15,16]. This will be of great interest for crops that provide organoleptic-sensitive products, such as grapevines, in which fine-tuned metabolic changes could lead to significant sensory impacts on grapes and the resultant wines [5,9]. Studies have examined the metabolic effects of pure-cultured fungal endophytes on grape cells under well-controlled conditions in dual-culture systems, in which certain degrees of fungal specificities have been shown to shape patterns in grape cellular metabolites [10,12]. At the same time, elicitors extracted from fungal or bacterial endophytes have been repeatedly shown to initiate various physiological and biochemical responses in host plants or cells [17,18,19]. Exposure to different EF strains created different metabolite patterns in grape cells (Figure 1 and Figure 2). Similarly, treatment with soluble extracts from different fungal strains also differentially modified the profiles of the cellular metabolites in grapes (Figure 1 and Figure 2). These results confirm the EF strain-dependent effects of both the living organism and the derived elicitors on cellular metabolites in grapes. Moreover, the specific impacts of fungal strains on metabolites in grape cells when separately exposed to different EF and CFEs (Figure 3 and Table 1) indicates that they can be used to regulate the biochemical qualities of grapes and wine with distinct characteristics. For example, endophytic fungal strain C11 and its extracts (C11E) can be purposely utilized to regulate metabolites in the compound class of amino acids, while fungi R12 and R12E conferred more effects on lipids, such as glycerol ester, lysophosphatidylcholine, and lysophosphatidyl ethanolamine, according to our results (Table 1). Amino acids are precursors of multiple secondary metabolites which are grape and wine biochemical and sensory quality contributors [20]. Metabolites belonging to the lipid class are substances which contribute important roles to the aroma of grapes and wines [21].

### 4.2. The Use of Different Forms of Regents Expanded the Functions of Endophytes in Crop Biochemical Quality Regulations

Several studies have been conducted on the impacts of endophytes and elicitors on host plants. This work, however, assessed metabolite profiles simultaneously in EF- and CFE-exposed grape cells, which allowed us to make a detailed comparison between the effects of certain fungi and their extracts on grape cells. The EF strains used in the experiments were from different fungal genera and are found in several varieties of grape and in vine-growing regions worldwide [5,22,23,24,25]. It is obviously important to elucidate the possible functions of these EF on grapevines for viticulture and vinification. The differentially initiated counts of SIMs (Table 1) in EF- and CFE-exposed grape cells represent differences in the intensity of effects (Figure 3). They are characterized by the qualitative differences in impacts between the use of fungi and fungal extracts. For example, C11 exposure triggered 5-aminovaleric acid, *N*-acetylaspartate, *N*-acetyl-L-glutamic acid, *N*-α-acetyl-L-arginine, *N*-(3-indolylacetyl)-L-alanine, leucylphenylalanine, and others in grape cells, but other SIMs were triggered in C11E-exposed grape cells (Table 1). Further, a phenolic acid class metabolite (3-hydroxy-4-isopropylbenzylalcohol 3-glucoside) was significantly promoted in C11E-exposed grape cells, but this metabolite was not significantly influenced in C11-exposed grape cells (Table 1). Accordingly, grape cells exposed to R12 triggered 20 specific SIMs, and in turn, seven specific SIMs were generated in R12E-treated grape cells. The exposure of grape cells to R32 initiated 16 specific SIMs, but R32E-treated grape cells only produced four specific SIMs (Figure 3 and Table 1). These specifically triggered SIMs, as compound classes of tannins, flavonoids, stilbene, phenolic acids, and organic acids, are major contributors to the sensory, biochemical, and protective qualities of grapes and wines [26].

In addition, DRMs in EF- and CFE-exposed grape cells indicated the differences in the effects between a fungus and its derived elicitors on plants (Figure 4). Among the 60 most commonly produced metabolites in this analysis, exposure to C11 and C11E produced nine DRMs in grape cells, including 2-isopropylmalate, nicotinate D-ribonucleoside, resveratrol-*O*-diglucoside, 1-*O*-[(E)-p-cumaroyl]-β-D-glucopyranose, 4-hydroxybenzaldehyde, gluconic acid, *N*-oleoylethanolamine, and others. In R32- and R32E-exposed grape cells, 11 DRMs were detected, including octadeca-11E,13E,15Z-trienoic acid, glucarate *O*-phosphoric acid, L-(+)-arginine, D-sedoheptuiose-7-phosphate, 2′-deoxyinosine-5′-monophosphate, and others. Only two DRMs (2-isopropylmalate and 3-hydroxy-3-methylpentane-1,5-dioic acid) were detected in R12- and R12E-treated grape cells (Figure 4). Many of these metabolites conferred important biochemical and organoleptic qualities into the grapes and wine contained in them [27]. In conclusion, the use of both EF and CFEs expanded the functions of endophytes in regulating the biochemical characteristics of crops. The results elucidated that the EF and its derived CFEs could be separately applied to emphasize those effects related to their specific SIMs and DRMs.

### 4.3. Majority of Metabolites Coordinately Responded in EF- and CFE-Exposed Grape Cells

As discussed above, living EF and fungal extracts produce different metabolic effects on grape cells. It is believed that the living organisms of endophytes and their elicitors have similar effects on host plants. It has been found that dual cultivation with fungal endophytes can quantitatively and compositionally modify anthocyanins in grape cells, and their fungal extracts have similar effects [16]. In this work, exposure to living fungi and fungal extracts resulted in different metabolite patterns in grape cells. However, underlying coordinative changes in relative contents of metabolites between EF- and CFE-exposed grape cells were observed. First, a large proportion of SIMs in grape cells treated with fungal extracts were co-detected simultaneously in the corresponding grape cells exposed to living fungi (Figure 3). Fungal strain-specific SIMs exhibited coordinative regulation in both living EF- and CFE-exposed grape cells. These SIMs included lysine butyrate in C11- and C11E-treated grape cells; eriodictyol 7-*O*-glucoside, lysoPC 18:0, lysoPE 16:0, lysoPE 18:2, and lysoPE 18:1 in R12- and R12E-exposed grape cells; and caffeine in R32- and R32E-exposed grape cells (Table 1). The significant differences in the RI values of metabolites between the EF- and CFE-exposed grape cells represented obvious coordinative metabolic responses. However, the degrees of coordinative change between the EF- and CFE-exposed grape cells appeared to exhibit an obvious dependence on fungal strain. R12 and R12E exposure conferred the largest degree of metabolite coordination in exposed grape cells, relative to the other two paired treatments used. Moreover, a certain proportion of CRMs identified in this assay (43.3% in paired treatments C11-/C11E- and R12-/R12E-exposed grape cells and 23.3% in R32-/R32E-treated grape cells, respectively) were detected in grape cells of different paired treatments, and the proportion of CRMs also appeared to show fungal strain dependencies (Figure 4). These CRMs covered a range of classes of metabolites, including piceid, 1-*O*-[(E)-p-cumaroyl]-β-D-glucopyranose, aminomalonic acid, pyridoxine, and lysoPC 18:1 (2n isomer), which are important in the sensory qualities of wine and in the beneficial effects on health. This also included key precursor substances for synthesizing other grapes and wine quality determination secondary metabolites (Victoriamoreno-Arribas and Carmenpolo 2009). In conclusion, with regard to the metabolome, obvious coordinative changes of metabolites between grape cells exposed to living EF and fungal extracts were observed. Certain proportions of fungal strain-specific CRMs were produced simultaneously in EF- and CFE-exposed plant cells. These results provided selectable applications of living fungi or their elicitors in regulating the biochemical quality of crops. The use of endophytic extracts has the advantage of having a well-controlled dosage and time of application, as well as avoiding the risk that the used fungi may become a pathogen.

## Figures and Tables

**Figure 1 molecules-27-05566-f001:**
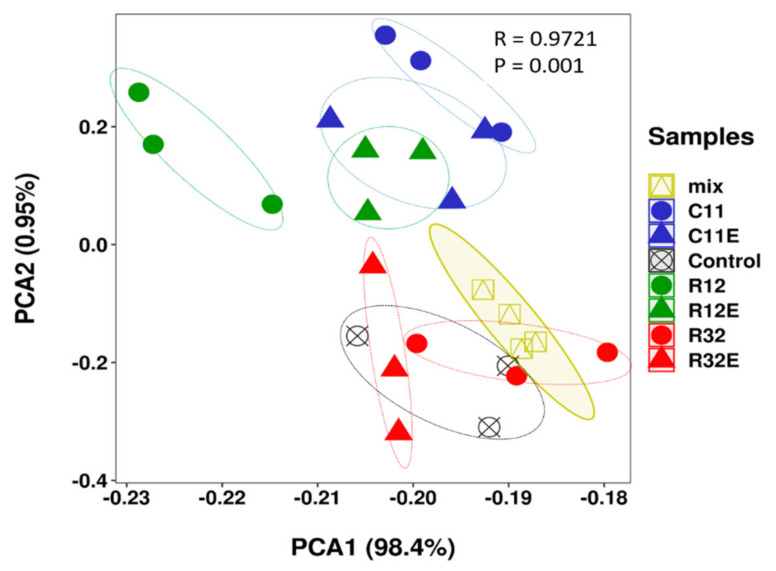
Principal component analysis (PCA) showing the overall metabolite patterns in grape cells of different treatments. The mixed sample group (mix) contains samples randomly uploaded during metabolite profiling for technical quality control. C11, R12, R32, C11E, R12E, and R32E briefly represent the sample groups treated with the corresponding endophytic fungal strains or their derived fungal extracts. Samples belonging to one treatment pair are exhibited as the same color in the plot.

**Figure 2 molecules-27-05566-f002:**
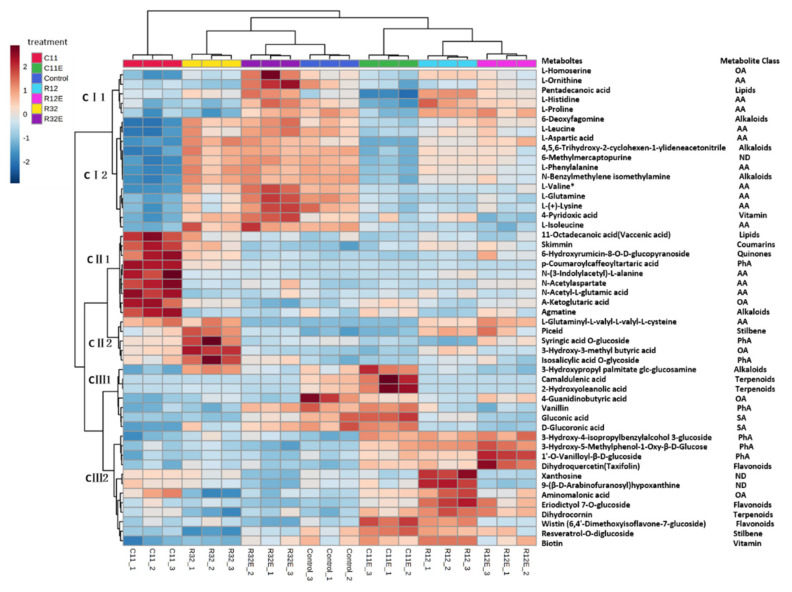
Heatmap and hierarchical cluster of the 50 most influenced metabolites in fungi- and fungal extract-exposed grape cells. C11, R12, R32, C11E, R12E, R32E, and control briefly represent the samples treated with the corresponding endophytic fungal strains or their derived fungal extracts or no treatment as control. The abbreviated metabolite classes are: OA: organic acids; AA: amino acids and derivatives; ND: nucleotides and derivatives; PhA: phenolic acids; and SA: saccharides and alcohols.

**Figure 3 molecules-27-05566-f003:**
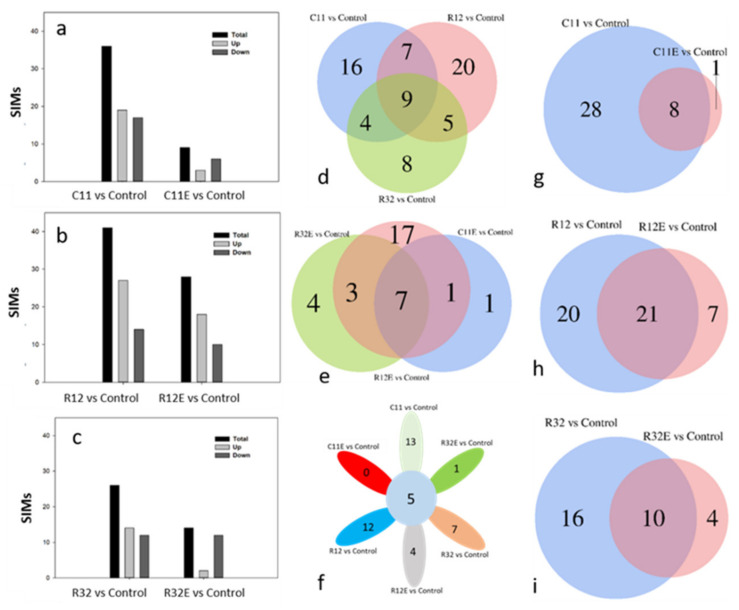
Comparing the number of significantly influenced metabolites (SIMs) initiated in endophytic fungi- and fungal extract-exposed grape cells. Bar plots (**a**–**c**) display the number of SIMs triggered in different EF- and CFE-exposed grape cells. The Venn plots (**d**–**i**) show the number of specific or co-initiated SIMs between or among different treatments. C11, R12, R32, C11E, R12E, and R32E briefly represent the sample groups treated with the corresponding endophytic fungal strains or their derived fungal extracts.

**Figure 4 molecules-27-05566-f004:**
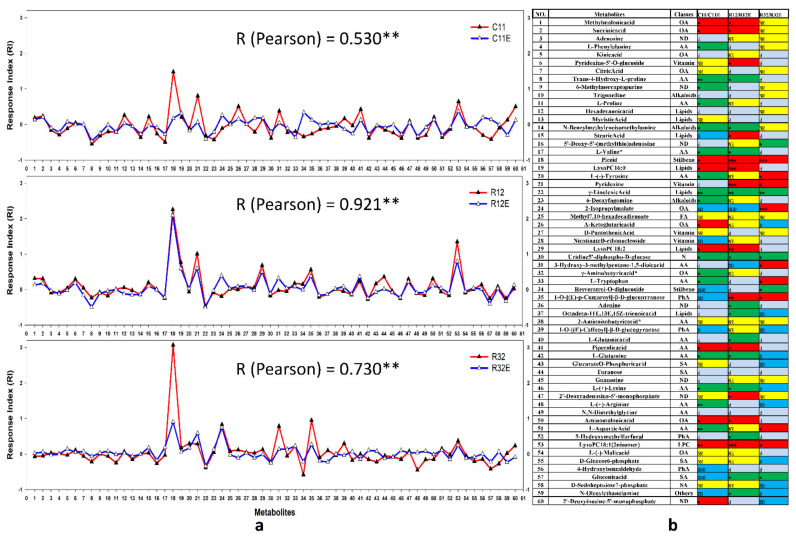
Coordinative responses of metabolites in grape cells exposed to endophytic fungi (EF) and corresponding fungal extracts (CFE). C11, R12, R32, C11E, R12E, and R32E briefly represent the sample groups treated with the corresponding endophytic fungal strains or their derived fungal extracts. The coordinative responses of the metabolites were analyzed according to the values of the response indexes (RIs), and the results show the coordinative responses of the top 60 metabolites with the most relevant contents (top 60 metabolites) that occupied more than 90% of the detected total metabolites in this experiment. (**a**) Correlations among metabolites in grape cells in each paired treatment, where ** indicates critically significant correlations at 0.01. (**b**) Response types of the top 60 metabolites in grape cells in each paired treatment: coordinately responded metabolites (CRMs, marked in red [upregulated, RI values of both treatments in the paired treatments (RIPT) more than or equal to 0.1] and green [downregulated, RIPT ≤ − 0.1]); differentially responded metabolites (DRMs, blue); almost non-responding metabolites (NR, in yellow); and other responding metabolites (d, in gray) in grape cells of each paired treatments. The degrees of coordination of the CRMs are represented by 1–3 stars: *, absolute RIPT greater than or equal to 0.1 and less than 0.3; **RIPT greater than or equal to 0.3 but less than 0.5; ***, RIPT greater than or equal to 0.5. DRMs are defined as the metabolites that were oppositely regulated in grape cells of the paired treatments, together with the absolute value of RI difference of the metabolites between the paired treatments greater or equal to 0.2. The degrees of different regulations of DRMs are given as variations of D: D indicates the absolute value in the difference of RIs in one treatment of the paired treatments (TA) and another treatment of the paired treatments (TB) greater than or equal to 0.2, but less than 0.3 (0.2 ≤ ∣TA − TB∣ < 0.3); DD indicates 0.3 < ∣TA − TB∣ ≤ 0.5; and DDD indicates ∣TA −TB∣ > 0.5. The yellow NR denotes metabolites in grape cells that were almost unresponsive to both treatments of the paired treatments, defined as an absolute RIPT less than 0.1. The abbreviations for the metabolite class names are as follows: OA, organic acids; ND, nucleotides and derivatives; AA, amino acids; PhA, phenolic acids; and SA, saccharides and alcohols.

**Table 1 molecules-27-05566-t001:** The significantly influenced metabolites (SIMs) detected in grape cells exposed to endophytic fungi and fungal extracts.

Metabolites Class (SIMs/TDM)	Metabolites	Significance Compared to the Control					
		C11	C1E	R12	R12E	R32	R32E
Alkaloids (1/12)	Caffeine	−1	−1	− *	−1	− ***	− *
Amino Acids and Derivatives (16/61)	5-Aminovaleric acid	− *	−1	−1	0	−1	0
	*Trans*-4-Hydroxy-L-proline	− *	−1	−1	−1	−1	0
	L-Tyramine	− *	− *	− *	− *	− *	− ***
	1,2-*N*-Methylpipecolic acid	− *	−1	−1	−1	−1	− *
	*N*-Acetyl-L-leucine	+*	+1	0	−1	+1	+1
	*N*-Acetylaspartate	+*	0	0	+1	0	+1
	*N*-Acetyl-L-glutamic acid	+*	0	+1	0	−1	0
	*N*-Acetylmethionine	− *	− *	− *	− *	− *	− **
	*N*-α-Acetyl-L-arginine	+*	0	0	0	0	1
	*N*-Acetyl-L-tyrosine	+*	+1	+1	0	+1	+1
	Lysine butyrate	− *	− *	−1	−1	0	0
	*N*-(3-Indolylacetyl)-L-alanine	+*	+1	+*	+1	+1	+1
	L-Homocystine	−1	0	− *	−1	0	−1
	Leucylphenylalanine	−1 *	0	− *	−1	0	0
	L-Glutamic acid *O*-glycoside	−1	0	−1	−1 *	−1 *	−1
	L-Glutaminyl-L-valyl-L-valyl-L-cysteine	+**	N/A	+**	+**	+**	N/A
Lipids (16/56)	γ-Linolenic acid	−1	−1	−1	− *	−1	−1
	11-Octadecanoic acid(Vaccenic acid)	+*	−1	+1	0	+1	+1
	13-HOTrE(r)	−1	−1	+*	0	0	−1
	1-α-Linolenoyl-glycerol	−1	−1	− *	−1	−1	−1
	LysoPC 18:1	+1	+1	+*	+1	+1	+1
	LysoPC 18:1(2n isomer)	+1	+1	+*	+1	+1	+1
	LysoPC 18:0	0	0	+*	+*	0	0
	LysoPC 18:0(2n isomer)	0	+1	+1	+*	0	0
	LysoPE 16:0	+1	+1	+*	+*	−1	+1
	LysoPE 16:0(2n isomer)	0	+1	+*	+*	0	+1
	LysoPE 18:3	−1	−1	+1	+1	− **	−1
	LysoPE 18:2	0	+1	+*	+*	−1	+1
	LysoPE 18:2(2n isomer)	+1	+1	+*	+*	0	1
	LysoPE 18:1	−1	+1	+*	+*	− **	+1
	LysoPE 18:1(2n isomer)	+1	+1	+*	+*	+1	+1
	Choline alfoscerate	−1	+1	+1	0	−1	− *
Nucleotides and Derivatives (3/40)	5-Methylcytosine	− *	0	−1	−1	−1	−1
	Xanthine	− *	−1	−1	−1	−1	−1
	9-(β-D-Arabinofuranosyl) hypoxanthine	+1	0	+*	−1	0	−1
Saccharides and Alcohols (3/24)	D-Glucoronic acid	− *	0	−1	− *	−1	−1
	D-(+)-Melezitose	+1	0	+*	+1	−1	+1
	D(+)-Melezitose *O*-rhamnoside	+*	+1	+*	+1	+1	+1
Vitamins (2/11)	Nicotinamide	−1	+1	+1	0	− *	−1
	Pyridoxine	+1	0	+*	+1	+1	+1
Organic acids (7/30)	2-Furanoic acid	+1	+1	0	−1	− *	0
	3-Hydroxy-3-methyl butyric acid	+1	0	+1	0	+*	0
	6-Aminocaproic acid	− *	−1	−1	−1	−1	0
	3-Hydroxyanthranilic acid	− **	0	+1	+1	−1	+1
	Diethyl phosphate	+1	0	− *	−1	0	−1
	3,4-Dihydroxybenzeneacetic acid	+*	+1	+*	+1	+1	+1
	*Trans*-4-Hydroxycinnamic acid methyl ester	−1	−1	− *	−1	−1	−1
Phenolic Acids (11/42)	Methyl ferulate	+*	0	+1	+*	+*	+*
	3-Hydroxy-5-Methylphenol-1-oxy-β-D-Glucose	0	+1	+1	+*	0	−1
	Isosalicylic acid *O*-glycoside	+1	0	+1	+1	+*	+1
	Feruloylmalic acid	+*	−1	+*	+*	+*	+*
	3-Hydroxy-4-isopropylbenzylalcohol 3-glucoside	0	+*	+*	+*	+1	+1
	1′-O-Vanilloyl-β-D-glucoside	−1	+1	+1	+*	0	−1
	Feruloyl glucose	−1	+1	+1	+1	+*	+1
	Syringic acid *O*-glucoside	+1	−1	− *	+1	+*	−1
	Trihydroxycinnamoylquinic acid	+*	−1	− **	−1	+*	− **
	Syringin	+1	0	+*	+*	+*	0
	p-Coumaroylcaffeoyltartaric acid	+*	+1	+1	0	+*	+1
Others (14/28)	Indole	− *	−1	0	−1	−1	0
	Piceid	+*	+1	+1 *	+*	+*	+1
	ε-Viniferin	+*	− **	+*	− **	0	− **
	Resveratrol-*O*-diglucoside	−1	+1	+1	0	− *	−1
	Camaldulenic acid	− ***	+1 *	− ***	− ***	− ***	− ***
	2-Hydroxyoleanolic acid	− *	+*	− *	− *	− *	− *
	6-Hydroxyrumicin-8-*O*-D-glucopyranoside	+*	−1	0	+1	+*	0
	Skimmin	+*	+1	+1	+1	+*	+1
	2-O-Galloyl-β-D-glucose	+1	−1	− *	+*	+*	− *
	Eriodictyol 7-*O*-glucoside	+1	+1	+*	+*	+1	−1
	Hesperetin 7-O-neohesperidoside (Neohesperidin)	− ***	− *	− **	− ***	−1	− ***
	Isorhamnetin-3-*O*-β-D-glucoside	0	+1	+*	+1	+1	0
	Octadecenoic amide	+1	−1	+*	0	+1	−1
	Propyl 2-(trimethylammonio)ethyl phosphate	+*	+1	+*	+1	+1	+1
	α-Viniferin	− ***	−1 *	+1 *	− ***	− *	− *

Note: C11, R12, R32, C11E, R12E, and R32E briefly represent the samples treated with the corresponding endophytic fungal strains and their derived fungal extracts; + and - represent metabolites being up- or downregulated, respectively, compared to the control. *: SIMs in treated grape cells. “*”: VIP ≥ 1 and satisfied the conditions as 1 ≤ ∣Log2(FC)∣ <5; “**”: VIP ≥ 1, 5 ≤ ∣Log2(FC)∣< 10; “***”: VIP ≥ 1, ∣Log2(FC)∣ ≥ 10; “0”: indicates that metabolites in the treated grape cells were not obviously influenced, defined as absolute Log2(FC) < 0.5; “1”: indicates ∣Log2(FC)∣ ≥ 0.5 but less than 1.0 meaning that the metabolites were obviously influenced in treated grape cells but did not reach statistical significance. N/A: the metabolites were not detected in the samples. The generated numbers of SIMs from each compound class and the total detected metabolite numbers of the classes are exhibited in the table.

## Data Availability

All data and materials are available in the manuscript.

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
