# Peer review of "Coordinative Changes in Metabolites in Grape Cells Exposed to Endophytic Fungi and Their Extracts"

_molecules, 2022, doi:10.3390/molecules27175566_

Round 1

Reviewer 1 Report

Abstracts are wordy and not clear, may redundant words need to be removed, and some results have to be stated.

Authors don’t follow the journal numerical reference style  

The introduction is very good

Material and methods is very clear

The result writing and figures quality is very poor, and I don’t understand it

I can't detect which is the fungal and which is the extract effect

A lot of abbv. And codes I can't follow all this

Figure 2 and 3 is very poor in quality and in explanation

I can't understand fig. 3

discussion is good

put conclusion with a single title and in a clear and concrete

Author Response

Dear reviewer

The authors are grateful for giving us valuable comments on our manuscript (molecules-1860993) entitled " Coordinative Changes in Metabolites Between Grape Cells Exposed to Endophytic Fungi and their Extracts ". All suggestions and comments are helpful to better improve our manuscript. After learned all those kindly raised comments, the manuscript was carefully revised accordingly and responded point to point as bellows.

Point 1: Abstracts are wordy and not clear, may redundant words need to be removed, and some results have to be stated.

Response to point 1: The abstract section has been revised as possible as we can to make it more refinedly and clearly. Redundant words have removed and some detailed results have added. Thank you!

Point 2: Authors don’t follow the journal numerical reference style

Response to point 2: References in the amnuscript were revised to follow the numerical style of the journal.

Point 3: The introduction is very good.

Response to point 3: Thank you!

Point 4: Material and methods is very clear

Response to point 4: Thank you!

Point 5: The result writing and figures quality is very poor, and I don’t understand it

Response to point 5: The results section has revised furtherly to make the part more clear and readable. In this vision, all figures were greatly promoted the qualities and some of the inappropriates whin the figures, figure’s captions and ligends have also revised accordingly to make them more understandable. Thank you to point out these comments for better improving the manuscript.

Point 6: I can't detect which is the fungal and which is the extract effect

Response to point 6: Sorry for not clearly convering our results, owing to some inappropriate writings in our previous manuscript. In present vision, sentences that may lead to ambiguities in the manuscript have thoroughly revised and polished, which may make it easier to understand, as possible as we can. In this study, the treatments of fungal strains were reprented as the fungal stain ID (such as C11, R12 and R32), and the treatments of corresponding fungal extracts were marked as the fungal stain ID +E (extracts) (such as C11E, R12E and R32E), for that could be easily figure out the effects from fungi or from fungal extracts..

Point 7: A lot of abbv. And codes I can't follow all this

Response to point 7: The necessary abbreviations used in the manuscript have been simplified; each abbv. has given a clear definition at where it firstly appeared; and all used abbv. were listed in the manuscript, and if necessary, it could move to anywhere appropriate in the manuscript.

Point 8: Figure 2 and 3 is very poor in quality and in explanation. I can't understand fig. 3

Response to point 8: Figures have greatly improved the qualities, and the figure captions and ligends also revised accordingly; we hope the revisions have make these figures more understandable.

Point 9: discussion is good

Response to point 9: Thank you!

Point 10: put conclusion with a single title and in a clear and concrete

Response to point 10: A clear and concrete conclusions have provided in the abstract for the study, and sub-conclusions also added in each part of the discussion.

Reviewer 2 Report

Qu et al. describe metabolite profiling in Vitis vinifera calli after treated with fungi and fungal extracts to simulate entdophytic microbes. Samples were analysed using targeted metabolic profiling and the data statistically evaluated using PCA and hierarchical clustering analysis.

Them manuscript is interesting and relevant. However, a major shortcoming of the manuscript is that no raw data are given (quantification of each metabolite for each sample). A table summarising that must be included as supplementary data. Without such a dataset the manuscript is incomplete and it is even not possible to review the manuscript in an appropriate way.

In addition, it would be useful to give also absolute date for at least the most important metabolites inn the main body of the manuscript. In Table 1 only significances are given. I completely agree that this is important. However, absolute changes are at least as important.

In addition, in Table 1 the names of the metabolites should be checked, for instance:

Trans-4-Hydroxy-L-proline   →   trans-4-Hydroxy-L-proline (trans should be italic)

1,2-N-Methylpipecolic acid (N should be written italic; this applies also for the other compounds)

γ-Linolenic Acid   →   γ-Linolenic acid

Trans-4-Hydroxycinnamic Acid Methyl Ester   →   trans-4-Hydroxycinnamic acid methyl ester (trans should be italic)

3-Hydroxy-5-Methylphenol-1-Oxy-β-D-Glucose     3-Hydroxy-5-methylphenol-1-oxy-β-D-glucoside

Author Response

Dear reviewer

The authors are grateful for giving us valuable comments on our manuscript (molecules-1860993) entitled " Coordinative Changes in Metabolites Between Grape Cells Exposed to Endophytic Fungi and their Extracts ". All suggestions and comments are helpful to better improve our manuscript. After learned all those kindly raised comments, the manuscript was carefully revised accordingly and responded point to point as bellows.

Point 1: Qu et al. describe metabolite profiling in Vitis vinifera calli after treated with fungi and fungal extracts to simulate entdophytic microbes. Samples were analysed using targeted metabolic profiling and the data statistically evaluated using PCA and hierarchical clustering analysis.

Them manuscript is interesting and relevant. However, a major shortcoming of the manuscript is that no raw data are given (quantification of each metabolite for each sample). A table summarising that must be included as supplementary data. Without such a dataset the manuscript is incomplete and it is even not possible to review the manuscript in an appropriate way.

Response to point 1: Thank you for pointing out the shortcoming of our manuscript, owing to our carelessness. In present version, all raw data containing the quantification of each metabolite for each sample were provided as supplementary data, the way as your kind suggested.

Point 2: In addition, it would be useful to give also absolute date for at least the most important metabolites inn the main body of the manuscript. In Table 1 only significances are given. I completely agree that this is important. However, absolute changes are at least as important.

Response to point 2: We totally agree the importance to provide the absolute changes of values related to the most important metabolites within the main body of the manuscript, that will definitely enchance the significance of the work. However, data aquired by the metabolome profiling were relative qualified other than the absolute ones, due to the limilation of the analysis approach. In addition, the majority of important metabolites which have changed differently in endophytic fungi and fungal extracts treated grape cells have exhibited within results of figure 2, figure 3 and figure 4, and a separate represent of those important metabolites will some redundant with the results now presented. Despite all this, some important metabolites in grape cells with obvious responses to endophytic fungi and fungal extracts were more emphasized in the present version. Thank you for the good suggestion.

Point 3: In addition, in Table 1 the names of the metabolites should be checked, for instance:

Trans-4-Hydroxy-L-proline   →   trans-4-Hydroxy-L-proline (trans should be italic)

1,2-N-Methylpipecolic acid (N should be written italic; this applies also for the other compounds)

γ-Linolenic Acid   →   γ-Linolenic acid

Trans-4-Hydroxycinnamic Acid Methyl Ester   →   trans-4-Hydroxycinnamic acid methyl ester (trans should be italic)

3-Hydroxy-5-Methylphenol-1-Oxy-β-D-Glucose   →   3-Hydroxy-5-methylphenol-1-oxy-β-D-glucoside

Response 1: All metabolite names appeared throughout the manuscript, figures and tables were all revised according to your kind comments. Thank you!  

Reviewer 3 Report

The paper describes the elicitation of metabolites in grapes mediated by three endophytic fungi and by their extracts as well as the putative impact of these metabolites on the biochemistry and sensory qualities of grapes and wines.  Some improvements should be done in the paper as:

The figures must be improved overall.  In Figs. 2 and 4B the metabolite names were impossible to read. 

Authors must include in Supplementary material the MS data of identified compounds used to generate the results and figures shown on the paper.

What can be said about the metabolites that were elicited in the grapes? What are your likely roles?  An in-depth discussion of the results obtained should be done.

Author Response

Dear reviewer

The authors are grateful for giving us valuable comments on our manuscript (molecules-1860993) entitled " Coordinative Changes in Metabolites Between Grape Cells Exposed to Endophytic Fungi and their Extracts ". All suggestions and comments are helpful to better improve our manuscript. After learned all those kindly raised comments, the manuscript was carefully revised accordingly and responded point to point as bellows.

Point 1: The paper describes the elicitation of metabolites in grapes mediated by three endophytic fungi and by their extracts as well as the putative impact of these metabolites on the biochemistry and sensory qualities of grapes and wines.  Some improvements should be done in the paper as:

The figures must be improved overall.  In Figs. 2 and 4B the metabolite names were impossible to read.

Response to point 1: All figures in the manuscript have greatly improved the qualities with higher resolutions; figure 1 has majorly revised by deleting the PLSD analysis, and maintained only the PCA plots, owing to some redudant of the two methods; figure captions and ligends also revised accordingly, hoping the revisions have make these figures more understandable. Thank you!

Point 2: Authors must include in Supplementary material the MS data of identified compounds used to generate the results and figures shown on the paper.

Response 2: Thank you for pointing out the shortcoming of our manuscript, owing to our carelessness. In present version, raw data containing all the identified compounds for each sample were provided as supplementary data (Table s1).

Point 3: What can be said about the metabolites that were elicited in the grapes? What are your likely roles?  An in-depth discussion of the results obtained should be done.

Response 3: Thank you for raising these kind commentsï¼›we have tried to improve the manuscript acoording to the comments as possible as we can. The achievements of our work are clarified the detailed responses in metabolites between grape cells exposed to endophytic fungi and fungal extracts, with more emphasized the significantly influenced metabolites (SIMs), coordinately responding metabolites (CRMs) and differentially respoding metabolites (DRMs), for the system (living fungal agents and the derived elicitors) can be more efficiently utilized in practice. Owing to the tremendous doccuments have covered the contributions of all detected biochemical components to the grape and wines, our present work minimized the further discussion of those elicited metabolites. Despite all this, some important metabolites in grape cells with obvious distinct and coordinative responses to endophytic fungi and fungal extracts were more emphasized in the present version, in results and discussion sections.

Round 2

Reviewer 1 Report

Thank you all and good luck the manuscript become much better